# Mental health and suicide risk among homosexual males in Bangladesh

**Muhammad Kamruzzaman Mozumder**[1]*, **Umme Habiba Jasmine**[2], **Md. Ashikul Haque**[3], **Shamsul Haque**[4]

1 Department of Clinical Psychology, University of Dhaka, Dhaka, Bangladesh, 2 University of the Witwatersrand, Johannesburg, South Africa, 3 International Organization for Migration, Cox's Bazar, Bangladesh, 4 Department of Psychology, Monash University Malaysia, Sunway, Malaysia

☉ These authors contributed equally to this work.
* mozumder@du.ac.bd

**Data Availability Statement:** Data available publicly at https://osf.io/jfw6e/ with support from the Open Science Framework.

**Funding:** The author(s) received no specific funding for this work.

## Abstract

### Background

Homosexual individuals are at high risk of suicide, but there is a shortage of data from developing countries to confirm this. Estimates on mental health and suicide risk among male homosexuals in Bangladesh are needed to generate awareness and to plan services accordingly.

### Method

We assessed mental health and suicidal behavior of 102 self-identified homosexual males from a community-based organization that works with the sexual minority population.

### Results

One-third of the participants (32.4%) had experienced attempted suicide, and almost half (47.1%) had a history of suicidal ideation and self-harm (40.2%). Compared to a heterosexual sample, homosexual males had poorer mental health as they scored higher on Beck Hopelessness Scale (Cohen's d = 0.29) and General Health Questionnaire (GHQ) (Cohen's d = 0.57). The results revealed positive correlations between self-harm, suicidal ideation and suicide attempt scores. Participants with history of suicide attempt, suicide ideation and self-harm reported worse general health, more social dysfunction, and severe depression than those without such history.

### Conclusion

Suicidality and mental health conditions among homosexual males in Bangladesh have appeared to be alarming. Given the concerns, we offer some recommendations for practitioners and social workers who are serving this population in Bangladesh.

**Competing interests:** The authors have declared that no competing interests exist.

# Introduction

Bangladesh is a densely populated (1140/square km) south Asian country with a population of 169.1 million [1]. The majority of the population is Muslim (88.4%) and aged between 15–49 years (54.9%), with a male-to-female ratio of 100.2 [1]. Male homosexuals are a hidden sexual minority group in Bangladesh, with a decade-old estimation of the homosexual population indicating a size between 21833 to 110581 [2].

Similar to many countries, homosexual individuals in Bangladesh are still tagged with many negative connotations, such as bad, immoral, criminal, mad, dirty, and the carriers of disease, especially HIV [3, 4]. Such stigma is intricately connected with derogatory and discriminatory behavior toward homosexual individuals [5]. Bangladesh being a Muslim-majority country, the disapproval of homosexual behavior is rooted in religious scriptures [Jami at-Tirmidhi, hadith 1456; Sahih Al-Bukhari, hadith 6834] and supported by the national penal codes. Keeping the British colonial track, Bangladesh penal code criminalizes any voluntary carnal intercourse between men and termed it as "against the order of nature" [Section 377; 6]. Despite the legal position, the government of Bangladesh is supporting activities towards improving the health and psychosocial care of homosexuals and other sexual minority groups, especially in connection to HIV/AIDS prevention program [see 7].

Living in stigmatized and discriminatory environment often causes psychological morbidity and emotional consequences among homosexual and other sexual minority groups [8–11]. Self-loathing is another aspect that contributes to increased psychological disturbances among homosexual people [12]. Suicide is seen as the most destructive outcome of such short- and long-term emotional disturbances [see 13].

Research has shown that sexual minority communities have higher rates of reported suicidal behavior compared to the general population [11, 13, 14]. This has been a major concern among researchers and activists working with sexual minority populations [see 15]. Surveys on homosexual communities as well as experts working with them, noted higher suicide risk in this population, especially among young homosexual individuals [16, 17]. A notable rate of suicide attempts (9.1%) has been observed among the young community sample of gender-nonconforming children aged 6 to 12 years [18]. King, Semlyen [19] conducted a systematic review of mental health, suicide, and self-harm research on homosexual and bisexual samples and found that homosexual males are at higher risk of developing depression, anxiety and attempted suicide compared to the general population.

Sandfort, de Graaf [20] reported higher rates of mood disorders (OR = 2.93), anxiety disorders (OR = 2.61), and substance use disorders (OR = 0.92) among homosexual males compared to heterosexual males when using 12 months prevalence. The odds ratio increased for all indicators when lifetime prevalence was compared between homosexual and heterosexual males [20]. Although poorer mental health of homosexual males has been consistently reported in the mainstream literature, some epidemiological surveys showed no significant difference in mental health between homosexual and heterosexual males [21, 22]. The relationship between suicidal ideation and mental health [23] has been found repeatedly, especially the relations between depression and suicidal ideation [13, 24, 25]. Relation between mental health and suicidal behavior has also been reported among the suicide decedents; homosexual males were found to have a higher rate of previous mental health diagnoses compared to heterosexual males [26].

Information on suicidal behavior among homosexual males in Bangladesh has appeared in several anecdotal reports [see 27]. However, the lack of empirical data regarding suicide and mental health of the member of this community hinders the initiation of strategic actions to address the issues. Published data on homosexual males in Bangladesh are generally limited

except for areas such as HIV and sexually transmitted diseases (STD). In recent years, there has been a significant amount of work done on mental health and social aspect of homosexual males in the country [28–32]. Hussain and Chowdhury [28] conducted a qualitative study and found that role conflicts, adverse life experiences, poor coping, lack of protective measures, negative societal perception, and relationship difficulties are the key contributors to suicidal behavior among homosexual males in Bangladesh.

### The current study

Although there has been ample research on mental health and suicidal behavior among homosexual populations in Western countries, there is limited data accumulated from this population in Bangladesh. There are several reasons for this. First, researchers have limited access to this population, especially due to secrecy around homosexual identity. Second, there is a general discouragement from society not to work with this population. NGO employees and other social workers are sporadically attacked by people who believe that homosexuality is unreligious. Finally, the topic is not on the research priority list for the country. Despite all those adversities, we aimed to conduct this study with the assistance of an NGO that works for sexual minority people in the country. We had three specific objectives. First, to see the prevalence of self-harm, suicidal ideation, and suicide attempt history among homosexual males in Bangladesh. Second, if the history of self-harm, suicidal ideation, suicide attempt and mental health among homosexual males were different from heterosexual males. Third, if homosexual males with a history of increased self-harm, suicide ideation and suicide attempt had more mental health impairments as compared to those with a history of reduced self-harm, suicidal ideation and suicide attempt.

## Materials and methods

### Participants

One hundred two self-identified homosexual males participated in the study. They were selected using purposive sampling procedure from a community-based organization that works with the sexual minority population in Dhaka, Bangladesh. To confirm an individual is a homosexual, we used the definition from Francoeur and Perper [33]. According to them, "the occurrence or existence of sexual attraction, interest and genitally intimate activity between an individual and other members of the same gender" is considered homosexual.

The participants' age ranged from 14 to 48 years, with a mean of 24.6 years (SD = 6.31). Sex work was reported as the primary source of income for 20.6% of the participants, while 32.4% had an additional source of income alongside sex work. 25.5% reported other sources, such as small business or paid employment, and a small portion (8.8%) reported that they had no regular income. Half of the participants (52.9%) completed high school education, one-fifth (22.5%) completed primary schooling, and another one-fifth (18.6%) went to college-level education; the remaining 5.9% had no formal education.

### Materials

A semi-structured survey questionnaire was used to collect socio-demographic and other psychological and contextual data. The socio-demographic and contextual questionnaire contained items on age, education, income, sexual practice and history of self-harm, suicidal thoughts and suicide attempt.

The Bangla version of the General Health Questionnaire [GHQ 28; 34] was used as a mental health indicator. With 28 items, the GHQ 28 assesses psychological disturbances with scores

on the full-scale and four of its subscales, namely somatic symptoms, anxiety and insomnia, social dysfunction and severe depression. Participants reported the presence of symptoms on a scale ranging from "less than usual" to "much more than usual" with corresponding scores of "0" to "3". With a score range of 0–84, higher scores on the GHQ 28 and its subscales indicate poorer mental health. This measure has been reported to have adequate psychometric properties, including Cronbach's alpha 0.95, test-retest reliability r = .70 and concurrent validity with the Middlesex Hospital Questionnaire (r = 0.55) [35, 36]. Adequate test-retest reliability (Spearman rho = 0.68) of the Bangla version of the GHQ 28 has been reported by Banoo [as cited in 37].

The Bangla version of the Beck Hopelessness Scale [BHS; 38] was used to assess hopelessness among the participants. This 20-item scale with dichotomous (Yes/No) response options is a widely used measure of hopelessness, which is a known associate of depression and suicidality [39, 40]. The BHS is an internally consistent (alpha 0.93) and valid (r = .74 with clinician rating; r = .63 with the Beck Depression Inventory) instrument [38]. The Bangla version of the BHS was translated by Uddin [see 41].

## Procedure

The participants were interviewed at the drop-in centers of an NGO working with male homosexuals. The participants were all self-identified as homosexuals. Face-to-face interviews were conducted individually by the second and third authors. Staff members of the NGO introduced the interviewers to the participants, which was necessary due to the secrecy of the homosexual identity and the sensitive nature of the data. Two-stage verbal consent process was used in selecting participants. In the first step, the NGO staff asked the prospective participants if they were interested in joining this study. Based on their approval, the staff sent the prospective participants to the interviewers, where final consent was sought. The interviewer informed the participants about the purpose, procedures, confidentiality, and voluntary nature of the study. Verbal informed consent was taken instead of written consent to protect the participants' identity [42] and to avoid legal ramifications [as homosexuality is prohibited by law; 6] in this sensitive research topic. This research was approved by the research Ethics Committee of the Department of Clinical Psychology, University of Dhaka (project number: RR-221101).

Thus, the principle of non-malfeasance to the research participants was ensured by choosing verbal consent over the generally preferred written consent. The interviews were conducted in a private location at the drop-in centers. No compensation was paid to avoid a possible breach of voluntary participation. The study was carried out in strict compliance with the universal ethical guidelines for research with humans that, include autonomy, benevolence, non-malfeasance and justice. When permission for data collection was sought to the NGO working with male homosexuals, they gave permission after scrutinizing the proposal and ensuring that the best interest of their service recipient (i.e., male homosexuals) was not violated in the research process.

## Data analysis

Descriptive and comparative analyses of data were performed using the software package called JASP version 0.11.1 [43]. Necessary assumptions (normality and homogeneity of variance) for parametric tests were checked where applicable. Several variables violated normality (Shapiro-Wilik test), while a few violated the assumption of equality of variance (Levene's test). Due to the well-reported robustness of t-test against the violation of normality [44, 45], the non-normality has been ignored in these analyses. However, the violation of equality of

variance was taken into consideration and adjusted t-values with Welch correction were reported.

## Results

### Prevalence of self-harm, suicidal ideation, suicide attempt history

High prevalence for the history of self-harm (40.2%), suicidal ideation (47.1%) and suicidal attempt (32.4%) was observed among the participants. Point-biserial correlation indicated a significant correlation (r = .61, p < .01) between suicidal ideation and suicidal attempt. Self-harm had significant positive relations with suicidal ideation (r = .35, p < .01) and suicidal attempt (r = .37, p < .01).

### Mental health: Homosexual versus heterosexual people

Scores on BHS and GHQ-28, along with its subscales, were compared with the mean scores obtained from a heterosexual sample (i.e., reference group) on the same measures presented in Mozumder [37] using one-sample t-tests (Fig 1). Homosexuals were found to be suffering from more hopelessness, [t (101) = 2.91, p < .01, MD = 1.39 CI (0.44–2.33), Cohen's d = 0.29] and poorer mental health (indicated by the GHQ 28 score) than heterosexuals, t (101) = 5.72, p < .01, MD = 8.81, CI (5.76–11.87), Cohen's d = 0.57]. The homosexuals also reported higher

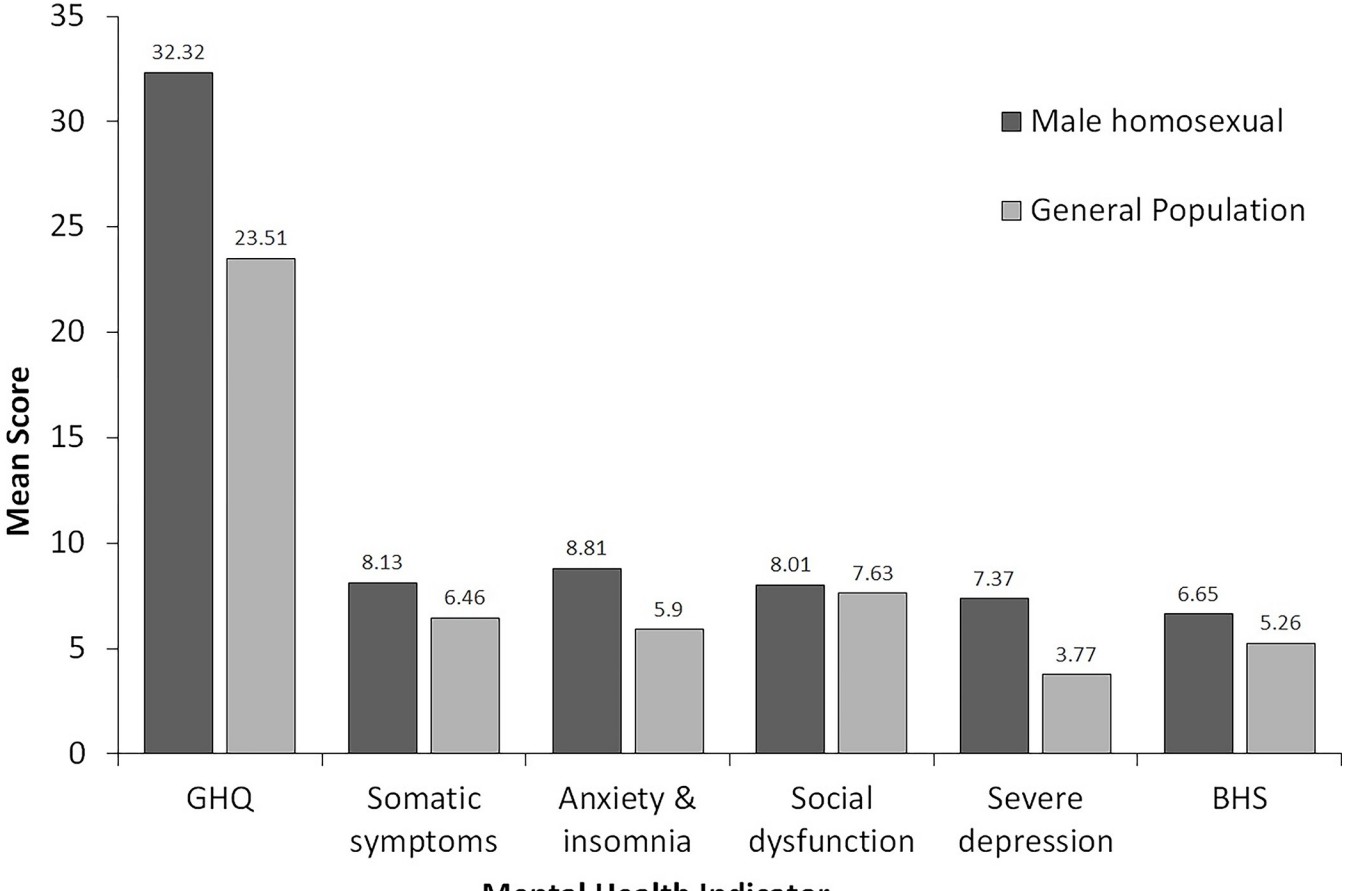

**Fig 1. Mean BHS and GHQ-28 scores for homosexual and heterosexual individuals.**

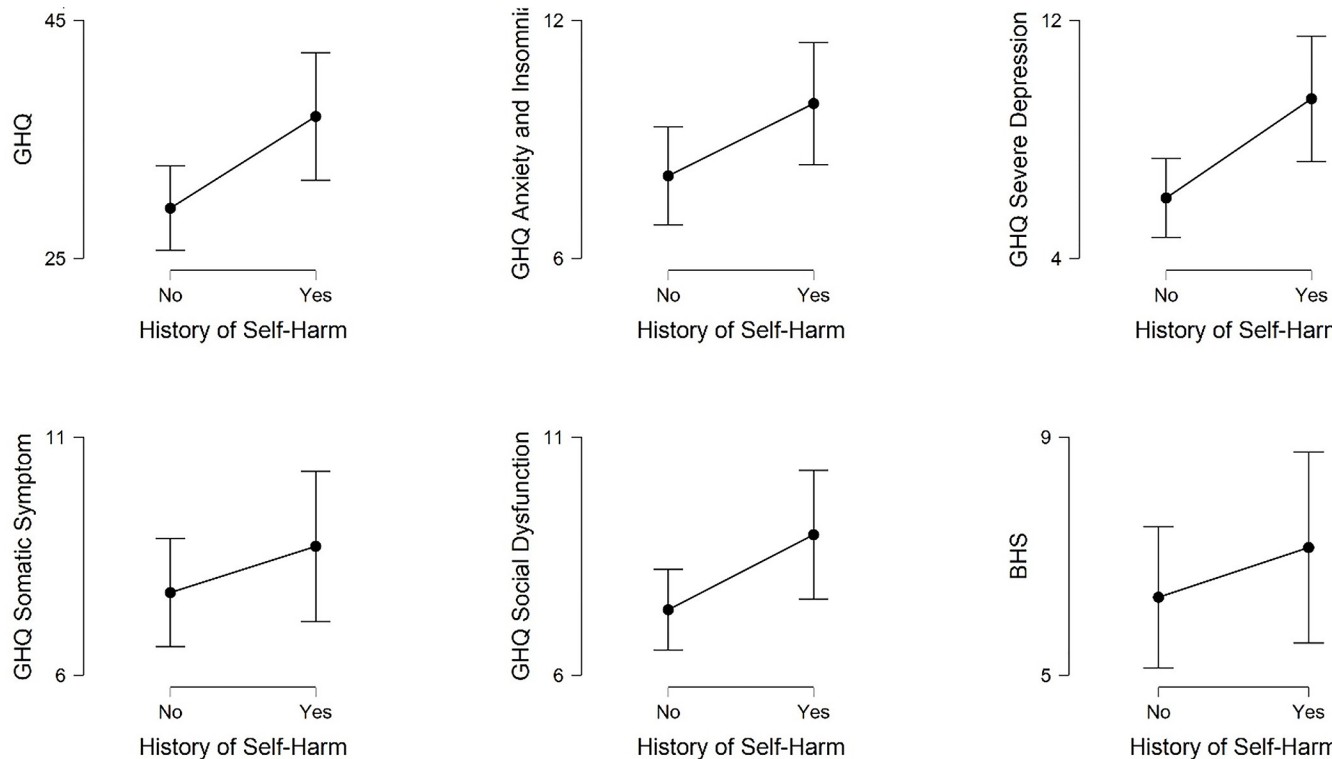

**Fig 2. Mean scores, with 95% confidence interval, on mental health indicators for male homosexuals with and without the history of self-harm.**

somatic symptoms [t (101) = 3.62, p < .01, MD = 1.67, CI (0.75–2.58), Cohen's d = 0.36], enhanced anxiety and insomnia [t (101) = 6.00, p < .01, MD = 2.91, CI (1.95–3.88), Cohen's d = 0.59], and severe depression than heterosexuals, [t (101) = 6.03, p < .01, MD = 3.60, CI (2.42–4.79), Cohen's d = 0.60]. However, the two groups were not different in terms of social dysfunction, [t (101) = 1.01, p > .05, MD = 0.38, CI (-0.36–1.12), Cohen's d = 0.10].

## Mental health of homosexuals with and without history of self-harm

We examined if mental health varied between homosexuals who had a history of self-harm (n = 41) and who had not (n = 61). A series of independent sample t-tests were performed and the results are presented in Fig 2. Those with a history of self-harm scored higher on the GHQ, meaning to have poorer overall mental health [t (100) = 2.51, p = .01, MD = 7.70, CI (1.62–13.78), Cohen's d = 0.51] than those without the history of self-harm. Compared to homosexuals who had no history of self-harm, those with such history demonstrated enhanced social dysfunction [t (70.88) = 1.99, p = .05, MD = 1.57, CI (.00–3.15), Cohen's d = 0.41], and severe depression [t (70.31) = 2.70, p < .01, MD = 3.33, CI (0.87–5.80), Cohen's d = 0.56]. However, the two groups did not show any difference in terms of somatic symptom [t (100) = 1.93, Cohen's d = 0.21], anxiety & insomnia [t (100) = 1.86, Cohen's d = 0.38], and hopelessness [t (100) = 0.86, Cohen's d = 0.17].

## Mental health of homosexuals with and without history of suicidal ideation

We compared homosexuals who had a history of suicidal ideation (n = 48) and those who had not (n = 54) in terms of different mental health indicators. A series of independent sample t-tests were performed and the results are presented in Fig 3. Those with a history of suicidal

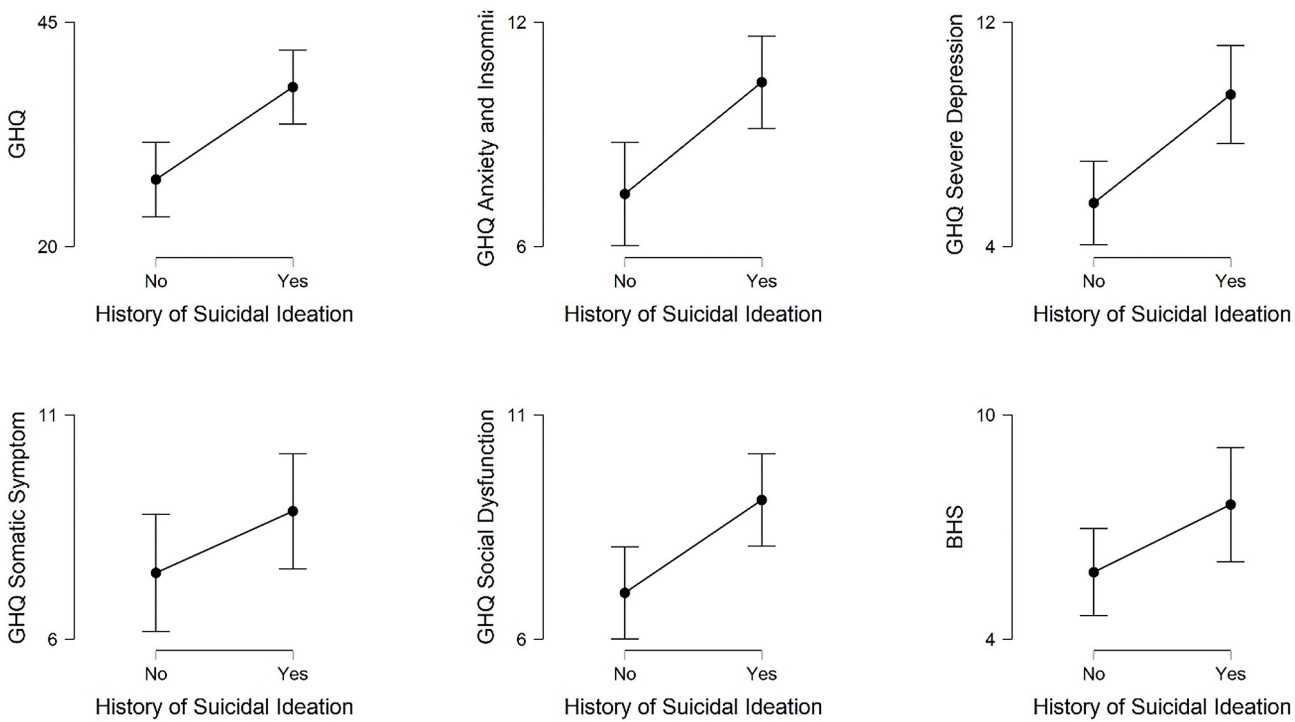

**Fig 3. Mean scores, with 95% confidence interval, on mental health indicators for male homosexuals with and without the history of suicidal ideation.**

ideation demonstrated higher scores on the GHQ, meaning to show poorer mental health compared to those without a history of suicidal ideation, [t (100) = 3.52, p < .01, MD = 10.29, CI (4.48–16.10), Cohen's d = .70]. They also reported enhanced anxiety and insomnia [t (100) = 3.21, p < .01, MD = 2.99, CI (1.14–4.84), Cohen's d = .64], social dysfunction [t (100) = 2.85, p < .01, MD = 2.07, CI (0.63–3.51), Cohen's d = .57], and severe depression [t (100) = 3.39, p < .01, MD = 3.86, CI (1.60–6.12), Cohen's d = 0.67] compared to those with suicidal ideation. However, the two groups were not different concerning somatic symptom [t (100) = 1.50, Cohen's d = 0.30] and hopelessness [t (100) = 1.90, Cohen's d = 0.38].

## Mental health of homosexuals with and without history of suicidal attempt

General mental health of homosexuals with (n = 33) and without (n = 69) the history of suicidal ideation was also compared. The results of independent sample t-tests revealed that except for hopelessness, those with a history of suicidal attempt had poorer mental health than those without such history (Fig 4). Homosexuals who had suicidal attempts scored higher on the GHQ, meaning they had poorer overall mental health [t (100) = 3.98, p < .01, MD = 12.24, CI (6.14–18.35), Cohen's d = 0.84], and enhanced somatic symptoms [t (100) = 1.98, p = .05, MD = 1.92, CI (0.00–3.84), Cohen's d = 4.12] than those without similar history. Compared to the non-attempted group, the attempted group also suffered more frequently from anxiety & insomnia [t (85.40) = 3.44, p < .01, MD = 3.05, CI (0.89–1.30), Cohen's d = 0.69], severe depression [t (51.59) = 3.98, p < .01, MD = 5.10, CI (1.29–2.52), Cohen's d = 0.88], and had social dysfunction [t (100) = 2.81, p < .01, MD = 2.18, CI (0.64–3.72), Cohen's d = 0.60].

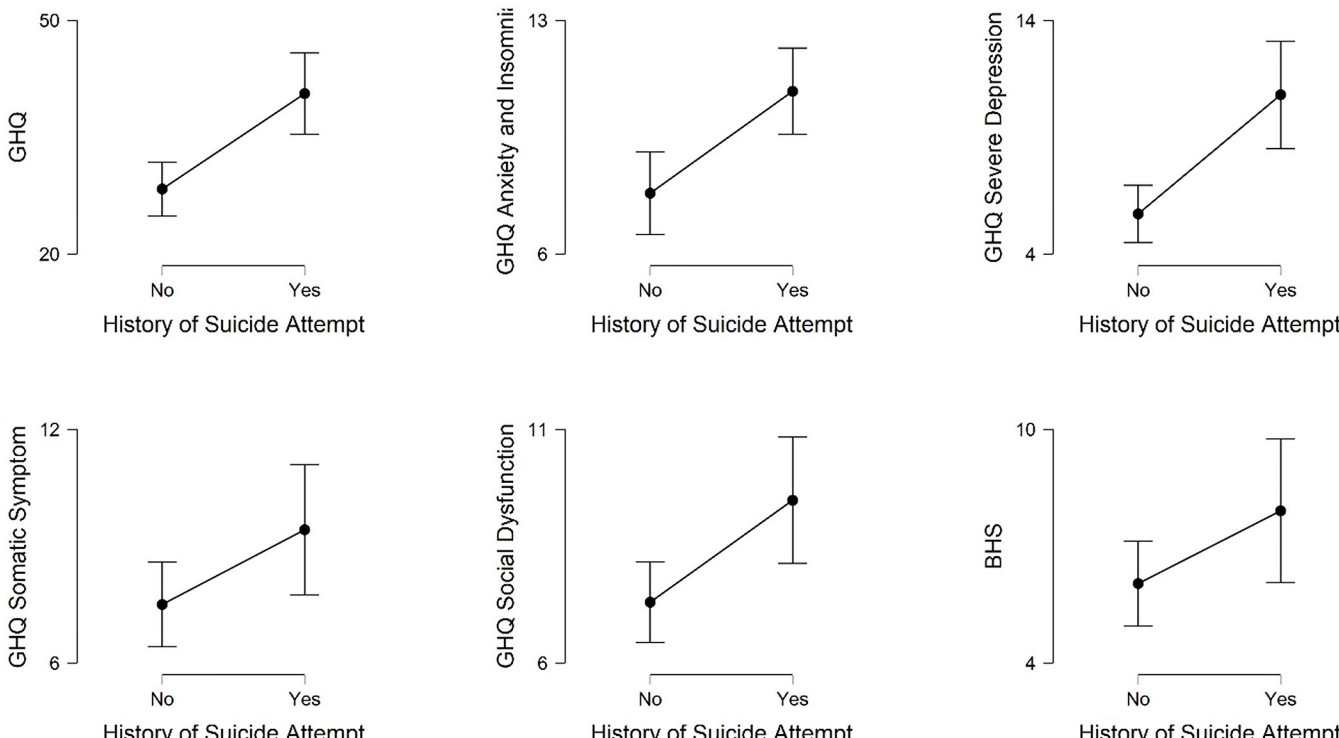

**Fig 4. Mean scores with 95% confidence interval on mental health indicators for male homosexuals with and without history of suicide attempt.**

## Discussion

The reports of increased rates of attempted and successful suicide among homosexual males in Bangladesh guided this research. The study had three specific objectives. First, to ascertain the rate of self-harm, suicidal ideation and suicide attempt among homosexual males. The findings showed that one-third (32.4%) of the participants had a history of attempted suicide and almost half (47.1%) had a history of suicidal ideation. These findings resemble the results from across the world, in which similar high rates of suicide attempts are reported among homosexual males [11, 46]. They have been reported to have a three- to four-fold higher risk of suicide attempts compared to heterosexual males [17, 19]. Although the lack of data from the general population will not allow us to draw such a conclusion from the present study, the reported prevalence is seemingly very high (32.4% and 47.1%) to render attention to the matter. As a single suicide attempt can bring fatal outcomes, half of our participants with suicidal ideation and one-third are at risk of suicide is obviously of serious concern.

History of self-harm was also high (40.2%) among the participants. Suicidal behavior and self-harm share common risk factors. A high concordance between self-harm, suicidal ideation and suicide attempt was observed, especially for suicidal ideation and attempt (r = .61, p < .01). Significantly higher rate of suicide attempts among the sexual minority population with suicidal ideation was also reported elsewhere [47].

Our second objective was to compare mental health of homosexual and heterosexual males. Scores on the BHS and GHQ 28 along with its subscales from the current sample were compared with data gathered earlier from a heterosexual sample published by Mozumder [37]. Results indicated significantly (p < .01) higher scores among homosexual males compared to the heterosexual sample in all indicators of mental health, except for the social dysfunction

subscale of the GHQ 28. The magnitude of these differences was assessed using Cohen's d values, which ranged from 0.29 to 0.60. The highest difference between homosexual males and heterosexual individuals was observed in the overall GHQ 28 scores, followed by the anxiety & insomnia and the severe depression subscales.

These results are indicative of poor mental health conditions among Bangladeshi homosexual males. They are also in line with the internationally available data on the mental health of homosexual males [11, 19, 20, 48]. Minority stress is one of the important predictors of poor mental health among this population [48]. Mental health services in Bangladesh are insufficient and burdened with inadequate professionals and limited accessibility, even for the general population. Homosexual identity presents itself as an additional accessibility barrier. Therefore, the majority of homosexual males are unlikely to seek services for their mental health needs. Relevant organizations would need to launch quality mental health services specifically designed to address the mental health needs of the sexual minority population.

The third objective was to see if mental health of the participants varied between those with and without a history of self-harm, suicidal ideation and suicide attempt. Hopelessness was one of the indicators of participants' mental health. Contrary to the common understanding that hopelessness and suicidal behavior are connected [47, 49, 50], no such connection between hopelessness and self-harm, suicidal ideation or suicide attempt was found. This suggests the possibility of exogenous causality of suicidal and self-harm behavior of homosexual males, as reflected in the exploratory study of Hussain and Chowdhury [28].

The overall GHQ 28 scores and most of its subscales were found to be related to self-harm, suicidal ideation or suicide attempt. Those who had a history of self-harm, suicidal ideation and suicide attempts scored significantly higher on overall GHQ 28 scores than those who did not have those histories (Cohen's d ranging from 0.51 to 0.84). Similar to the overall GHQ, the social dysfunction and severe depression subscales also significantly differentiated those who had and did not have these histories (Cohen's d ranging from 0.41 to 0.88). Scores on the anxiety & insomnia subscale of the GHQ 28 significantly differentiated individuals with and without a history of suicidal ideation (Cohen's d = 0.64) and suicide attempt (Cohen's d = 0.69). The somatic symptoms subscale was able to differentiate history and no-history only in the case of suicide attempt but marginally (t = 1.98, p = .05). Cohen's d improved in all the indicators in a linear manner from self-harm to suicidal ideation to suicide attempt. This suggests a possibility that mental health state was more closely associated with suicide attempts than with suicidal ideation and self-harm. The finding from the present study are in line with but more comprehensive compared to the existing literature on the relation between mental health and suicidal behavior [13, 23, 25, 47].

## Strengths and limitations

This study presents descriptive results on the relationship between mental health and suicidal behavior among homosexual males in Bangladesh. The results confirm the anecdotal reports that homosexual individuals are prone to suicidal behavior and more likely to have poorer mental health than heterosexual individuals. The mental health professionals and activists working with the sexual minority population may persuade the concerned authorities to take preventive measures in safeguarding the wellbeing of this vulnerable group of people. Since the main reasons for impaired mental health among homosexual people are the societal rejection and the hatred exerted towards them, the Govt. and the NGOs should work at the community level to change people's attitudes towards homosexuality. Religious leaders should also come forward to let the public know that all humans are created equal by the Almighty Allah, and everyone has the right to live with dignity.

This study has several limitations. First, we recruited a smaller sample and the sampling was non-random, posing a significant challenge to generalize our findings. Although we claim that purposive sampling used in this study is the most feasible choice considering the secretive nature of the participant group, one should be cautious about applying the results to the larger population. Second, the participants were all self-identified homosexual males. However, many of them had heterosexual exposure too, as they had wife with whom they were sexually engaged. Despite this fact, they still did not identify themselves as bisexual. As people shun homosexual individuals, consider them immoral and abnormal, and view homosexuality as a crime in Bangladeshi society, many homosexual individuals get involved in a heterosexual marriage only to satisfy their families and comply with societal expectations despite their lack of sexual interest towards the heterosexual partner.

## Conclusion

The mental health condition among homosexual males in Bangladesh appeared to be alarming. They have the tendency to commit suicide and make suicidal attempt more than heterosexual individuals. They are hopeless about their life, often deprived of family support and love. The results of the current study ring the bell for both Govt. and non-Govt. Organizations to seriously consider the issue and formulate policies to safeguard the wellbeing of this sexual minority population in the country. Politicians, community leaders and religious organizations should put hands together to understand the needs of these people and prepare the countrymen so that they can accept them with warm hearts. Mental health practitioners and social workers should work together to develop comprehensive prevention programs, particularly tailored for this population.

## Supporting information

**S1 File. Inclusivity in global research.**
(PDF)

## Author Contributions

**Conceptualization:** Muhammad Kamruzzaman Mozumder, Umme Habiba Jasmine, Md. Ashikul Haque.

**Data curation:** Muhammad Kamruzzaman Mozumder, Umme Habiba Jasmine, Md. Ashikul Haque.

**Formal analysis:** Muhammad Kamruzzaman Mozumder, Shamsul Haque.

**Investigation:** Muhammad Kamruzzaman Mozumder, Umme Habiba Jasmine, Md. Ashikul Haque.

**Methodology:** Muhammad Kamruzzaman Mozumder, Umme Habiba Jasmine, Md. Ashikul Haque.

**Project administration:** Muhammad Kamruzzaman Mozumder, Umme Habiba Jasmine, Md. Ashikul Haque.

**Supervision:** Muhammad Kamruzzaman Mozumder.

**Visualization:** Shamsul Haque.

**Writing – original draft:** Muhammad Kamruzzaman Mozumder.

**Writing – review & editing:** Muhammad Kamruzzaman Mozumder, Shamsul Haque.

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
