## [Decision Letter · Decision Letter 0]

22 Mar 2023

PONE-D-23-01885Mental Health and Suicide Risk among Homosexual Males in BangladeshPLOS ONE

Dear Dr. Mozumder,

Thank you for submitting your manuscript to PLOS ONE. After careful consideration, we feel that it has merit but does not fully meet PLOS ONE’s publication criteria as it currently stands. Therefore, we invite you to submit a revised version of the manuscript that addresses the points raised during the review process.

We look forward to receiving your revised manuscript.

Kind regards,

Sadia Jabeen, Ph.D.

Academic Editor

PLOS ONE

3. In the ethics statement in the Methods, you have specified that verbal consent was obtained. Please provide additional details regarding how this consent was documented and witnessed, and state whether this was approved by the IRB

Reviewers' comments:

Reviewer's Responses to Questions

**Comments to the Author**

1. Is the manuscript technically sound, and do the data support the conclusions?

Reviewer #1: Yes

2. Has the statistical analysis been performed appropriately and rigorously? 

Reviewer #1: Yes

3. Have the authors made all data underlying the findings in their manuscript fully available?

Reviewer #1: Yes

4. Is the manuscript presented in an intelligible fashion and written in standard English?

Reviewer #1: Yes

5. Review Comments to the Author

Reviewer #1: Summary of research and reviewer comment

The author has raised a very important issue in the paper and the overall impression of research paper is good. Gender diversity and sexual orientation are complex and often taboo topics in Bangladesh. The author has shown some audacity while deciding investigating this sensitive yet critical social phenomenon.

Minor discussion

-While the author talk’s suicide cases among homosexual, I really think the broader scenario of issue need to be discussed in introduction and literature review. For instance its prevalence in developed world, than developing and under developed world. In addition to that while you talk of Bangladesh, it is important for international reader to give an overall view of your country. A little literature on country’s geographical location, socio economic indicators and population dynamics will help understanding the presence of certain social phenomenon in country.

-The parental role and investigation from respondents about the way the respondents have been raised by parent’s was a big missing part in research.

-Similary, It would have been great if reaction was investigated from respondents about their thoughts and parents own perception while knowing their children are homosexual and that effect on their mental health.

-A literature could also cover the legality of such sensitive issue in Bangladesh as an Islamic state particularly or anything in pipeline by state while addressing homosexuality in country.

-The author can avoid references that are too old, like one mentions year 1984 and so.

6. PLOS authors have the option to publish the peer review history of their article (what does this mean?). If published, this will include your full peer review and any attached files.

Reviewer #1: **Yes: **SONIA OMER

---

## [Author Response · Author response to Decision Letter 0]

24 Mar 2023

1. Response to Reviewer. 

Summary of research and reviewer comment

The author has raised a very important issue in the paper and the overall impression of research paper is good. Gender diversity and sexual orientation are complex and often taboo topics in Bangladesh. The author has shown some audacity while deciding investigating this sensitive yet critical social phenomenon.

Response: Thank you for agreeing to review this article and making these comments to improve the quality of the article. 

Minor discussion

-While the author talk’s suicide cases among homosexual, I really think the broader scenario of issue need to be discussed in introduction and literature review. For instance its prevalence in developed world, than developing and under developed world. In addition to that while you talk of Bangladesh, it is important for international reader to give an overall view of your country. A little literature on country’s geographical location, socio economic indicators and population dynamics will help understanding the presence of certain social phenomenon in country.

Response: Thank you for these suggestions. We have included a brief summary of contextual details on Bangladesh in the introduction. However, we did not include a comparative analysis of suicide rates in developed, developing and underdeveloped worlds. Our initial planning had that in mind, but as there are many factors associated with the rates of suicide in a specific context, we felt it would require a carefully planned meta-analysis to make a meaningful comparison in that regard. We, therefore, avoided such premature misleading comparisons, which we could achieve through a brief literature review. 

Added section: 

Bangladesh is a densely populated (1140/square km) south Asian country with a population of 169.1 million [1]. The majority of the population is Muslim (88.4%) and aged between 15-49 years (54.9%), with a male-to-female ratio of 100.2 [1]. Male homosexuals are a hidden sexual minority group in Bangladesh, with a decade-old estimation of the homosexual population indicating a size between 21833 to 110581 [2].

-The parental role and investigation from respondents about the way the respondents have been raised by parent’s was a big missing part in research.

Response: Yes, that would add a wider perspective in interpreting the data; however, we did not collect data on the parental role. As the data collection is already completed, we may only try to accommodate that in future research. 

-Similary, It would have been great if reaction was investigated from respondents about their thoughts and parents own perception while knowing their children are homosexual and that effect on their mental health.

Response: This is another area that we cannot address now, as the data collection is already completed, we will try to accommodate this in future research. 

-A literature could also cover the legality of such sensitive issue in Bangladesh as an Islamic state particularly or anything in pipeline by state while addressing homosexuality in country.

Response: We have revised the writing and added a sentence to address this. 

Revised and added section:

Bangladesh being a Muslim-majority country, the disapproval of homosexual behavior is rooted in religious scriptures [Jami at-Tirmidhi, hadith 1456; Sahih Al-Bukhari, hadith 6834] and supported by the national penal codes. Keeping the British colonial track, Bangladesh penal code criminalizes any voluntary carnal intercourse between men and termed it as “against the order of nature” [Section 377; 6]. Despite the legal position, the government of Bangladesh is supporting activities towards improving the health and psychosocial care of homosexuals and other sexual minority groups, especially in connection to HIV/AIDS prevention program [see 7]

-The author can avoid references that are too old, like one mentions year 1984 and so.

Response: Thank you for this suggestion. We have checked the document thoroughly and removed a few old citations while adding a few new citations. Subsequent changes in reference sections have also been made. 

Removed: 

Herek (1984); Stein and Cohen (1984); Kourany, (1987); Robinson and Price (1882); Beck, et al (1990); Beck and Steer (1988)

Added: 

Bangladesh Bureau of Statistics, (2021); icddrb and Government of Bangladesh, (2012); Margolin, (2023); National AIDS/STD Programme (NASP) (2016); Kanbur, (2020); Taghavi, (2002); Liu et al., (2020); Baryshnikov et al., (2020);

 

2. Response to Academic Editor. 

Response: Thank you for this suggestion, we had missed a few in the last submission. Now we have revised the manuscript as per PLOS ONE style requirements. 

Response: We have completed the questionnaire and attached that as supporting information. 

3. In the ethics statement in the Methods, you have specified that verbal consent was obtained. Please provide additional details regarding how this consent was documented and witnessed, and state whether this was approved by the IRB

Response: We have revised the secion as per the suggestion

Revised and added section:

Two-stage verbal consent process was used in selecting participants. In the first step, the NGO staff asked the prospective participants if they were interested in joining this study. Based on their approval, the staff sent the prospective participants to the interviewers, where final consent was sought. The interviewer informed the participants about the purpose, procedures, confidentiality, and voluntary nature of the study. Verbal informed consent was taken instead of written consent to protect the participants’ identity [42] and to avoid legal ramifications [as homosexuality is prohibited by law; 6] in this sensitive research topic. This research was approved by the research Ethics Committee of the Department of Clinical Psychology, University of Dhaka (project number: RR-221101). 

Response: We have revised the statement as per the suggestion

Revised IRB statement:

This research was approved by the research Ethics Committee of the Department of Clinical Psychology, University of Dhaka (project number: RR-221101). 

Response: In line with the suggestions from the reviewer, we have revised the reference section, where 6 very old references were removed, and 8 references were added. 

Removed: 

Herek (1984); 

Stein and Cohen (1984); 

Kourany, (1987); 

Robinson and Price (1882); 

Beck, et al (1990); 

Beck and Steer (1988)

 Added: 

Bangladesh Bureau of Statistics, (2021); 

icddrb and Government of Bangladesh, (2012); 

Margolin, (2023); 

National AIDS/STD Programme (NASP) (2016);

Kanbur, (2020); 

Taghavi, (2002);

Liu et al., (2020); 

Baryshnikov et al., (2020);

---

## [Editor Report · Decision Letter 1]

24 Jul 2023

Mental Health and Suicide Risk among Homosexual Males in Bangladesh

PONE-D-23-01885R1

Dear Dr. Mozumder,

We’re pleased to inform you that your manuscript has been judged scientifically suitable for publication and will be formally accepted for publication once it meets all outstanding technical requirements.

Kind regards,

Sadia Jabeen, Ph.D.

Academic Editor

PLOS ONE
---

## [Editor Report · Acceptance letter]

2 Aug 2023

PONE-D-23-01885R1 

Mental Health and Suicide Risk among Homosexual Males in Bangladesh 

Dear Dr. Mozumder:

I'm pleased to inform you that your manuscript has been deemed suitable for publication in PLOS ONE. Congratulations! Your manuscript is now with our production department. 

Kind regards, 

on behalf of

Dr. Sadia Jabeen 

Academic Editor

PLOS ONE